# Combined Nano Silver, α-Aminoisobutyric Acid, and 1-Methylcyclopropene Treatment Delays the Senescence of Cut Roses with Different Ethylene Sensitivities

**Suong Tuyet Thi Ha and Byung-Chun In ***

Division of Horticulture and Medicinal Plant, Andong National University, Andong 36729, Korea; tuyetsuongha@gmail.com
* Correspondence: bcin@anu.ac.kr

**Abstract:** Flower senescence varies among cut roses (*Rosa hybrida* L.), and it is known that the postharvest life of ethylene-sensitive flowers is strongly related to the transcriptional accumulation of ethylene biosynthesis genes, whereas that of ethylene-insensitive flowers is shortened by water stress. These different responses of flowers to hormone and water stresses limit the action of preservatives in inhibiting the postharvest deterioration of cut rose flowers. This study was conducted to investigate the effects of the combined application of antibacterial agents and ethylene biosynthesis and binding inhibitors on the postharvest life and quality of the cut rose cultivars 'Matador' (ethylene-sensitive) and 'Dolcetto' (ethylene-insensitive). Cut flowers were treated with nano silver (NS), a combination of NS and α-aminoisobutyric acid (NS+A), or a combination of NS+A and 1-methylcyclopropene (NS+AM), and they were subsequently exposed to ethylene for 24 h. Treatment effectiveness was compared with that of control (CON) flowers, which were kept in distilled water and exposed only to ethylene. The results showed that all treatments significantly improved the postharvest quality and vase life of both rose cultivars. However, NS+AM most markedly delayed senescence and prolonged the vase life of cut flowers by 217% in 'Matador' and 168.4% in 'Dolcetto', compared with those of CON flowers. NS+AM also effectively increased the flower diameter and maintenance time of positive water balance and initial fresh weight by enhancing solution absorption as well as suppressing bacterial growth at the cut stem ends of the cut roses. Furthermore, NS+AM greatly suppressed the ethylene-inducible increase of ethylene biosynthesis genes and the reduction of ethylene receptor genes in petals, which resulted in a reduced flower response to exogenous ethylene in both rose cultivars. These findings show that NS+AM effectively delays flower senescence in both ethylene sensitive and insensitive cultivars by synergistically preventing ethylene response and water stress in cut flowers.

**Keywords:** cut flowers; ethylene inhibitors; ethylene response; nano particles; vase life; water relations

## 1. Introduction

Roses are one of the most important ornamental plants worldwide. However, they have a limited lifespan, and postharvest senescence often appears within a week for most cultivars when used as cut flowers [1,2], which is major challenge to their marketing. Early senescence symptoms in cut roses are closely related to increased hydraulic resistance caused by microbial growth in the vase solution, which causes physical occlusion xylem vessels [3,4]. The low water absorption and transport rates in the cut flower stems result in physiological disorders, such as bent neck (bending of peduncle) and the wilting of leaves and flowers, which thus shortens the vase life of the flowers.

In addition to the negative correlation between vase life and water stress, cut roses are sensitive to the plant hormone, ethylene. Early senescence symptoms in ethylene-sensitive rose cultivars are caused by the synthesis of ethylene in petals in relation to transcriptional

regulation of two ethylene biosynthesis genes, 1-aminocyclopropane-1 carboxylic acid synthesis (ACS) and 1-aminocyclopropane-1 carboxylic acid oxidase (ACO) [1]. In addition, exposure to exogenous ethylene can result in floral organ wilting or abscission, which decreases the postharvest life of cut rose flowers [1,2,5,6]. However, ethylene sensitivity varies widely among rose cultivars. Ethylene-sensitive cultivars show early senescence symptoms that are strongly related to increased ethylene biosynthesis gene mRNA levels, whereas early senescence symptoms in ethylene-insensitive cultivars are associated with a water relation failure [1]. These varied causes of early senescence make it difficult to manage the postharvest quality of cut roses and extend their postharvest life.

Numerous preservatives, such as silver thiosulfate (STS), have been studied and applied to reduce postharvest disorders and enhance the vase life of cut flowers [7–10]. STS has a strong antagonistic effect on ethylene binding and antibacterial activity, and it has been widely used to retard senescence and extend the postharvest life of cut flowers [7,10]. However, it contains heavy metals, which are potent environmental pollutants and are detrimental to human health. It is thus necessary to develop alternative preservatives that can be applied to a wide range of flower species and varieties in the postharvest industry. This could be achieved by combining certain chemical agents to develop a universal preservative that simultaneously suppresses microbial proliferation and ethylene damage. In this respect, α-aminoisobutyric acid (AIB) has been found to retard petal senescence and extend the vase life of various cut flower species by suppressing the expression of two ethylene biosynthesis genes, *ACS* and *ACO*, in flower petals [11–13]. However, the effect of AIB on postharvest quality of cut roses has not been well studied. In addition, 1-methylcyclopropene (1-MCP) is an inhibitor of ethylene action that has been found to improve the vase life and postharvest quality of various flower species including roses [9,13,14]. Previous observations showed that 1-MCP has a greater affinity for the ethylene receptors than ethylene [15]; therefore, 1-MCP effectively suppresses ethylene binding to the receptors and blocks the action of downstream components in ethylene signal transduction within plants. In addition to the blocking ethylene action, 1-MCP also affects ethylene biosynthesis in several plants through feedback inhibition [15]. Furthermore, nano silver (NS) has strong antimicrobial activity owing to its small practical size [16,17]. There has been recent interest in applying NS to prolong the postharvest life of cut flowers, and its positive effect on extending vase life in many flower species, such as gerbera, roses, carnations, and cosmos, has been reported [18–21]. Previous studies have indicated that NS prolongs the vase life of cut flowers by suppressing expression of ethylene biosynthesis and senescence-related genes and inhibiting the proliferation of bacteria in vase solution [18,19,21], and it has also been shown to decrease *Botrytis cinerea* growth on rose petals of cut roses by down-regulating virulence factor gene and ethylene-related genes [19]. However, to date, few studies have focused on determining how NS improves postharvest quality and vase life of cut rose flowers. Moreover, no studies have investigated the combined effects of using NS with ethylene biosynthesis and action inhibitors to improve the postharvest quality and longevity of cut flowers.

Therefore, in this study, we hypothesized that combined application of an antibacterial agent and ethylene synthesis and binding antagonists would improve the hydraulic conductance of cut stems and simultaneously suppress flower response to ethylene, thereby greatly extending the vase life of cut roses, regardless of their ethylene sensitivity. To test this hypothesis, two cut rose cultivars with different sensitivities to ethylene were treated with NS alone or with a combination of NS, AIB, and 1-MCP, and then, subsequently, exposed to ethylene. The physiological and morphological characteristics of the cut flowers, including water relations and petal senescence, were observed during the vase life period to determine the effects of the treatments. The mRNA levels of genes involved ethylene biosynthesis and their receptors were also monitored to test whether the combined use of AIB and 1-MCP simultaneously blocks ethylene binding and associated responses to it.

## 2. Material and Method

### 2.1. Plant Materials

In accordance with their use in our previous study [1], *Rosa hybrida* 'Matador' and 'Dolcetto' were selected for use in the experiment as ethylene-sensitive and ethylene-insensitive flowers, respectively. Rose plants were grown at a commercial greenhouse in Jeonju, Korea, where they were drip-irrigated with a nutrient solution containing nitrogen, potassium, calcium, magnesium, phosphorus, and other trace substances. Changes in environmental conditions (air temperature, relative humidity (RH), and light density) were monitored during the year in the greenhouse using data loggers. Rose flowers of the two cultivars were harvested at commercial maturity stage (loose pointed bud cylindrical) and immediately placed in buckets containing tap water. They were then transported to the laboratory within 4 h, where the flower stems were quickly trimmed with scissors in clean water to a length of 45 cm and including three upper leaves. The cut flowers were then held in a controlled environment room at $25 \pm 1\,^\circ\text{C}$ and at a RH of 50–55% prior to conducting treatments.

### 2.2. NS, AIB, and 1-MCP Treatments

To examine the combined effectiveness of using antibacterial agents and ethylene inhibitors on the vase life and postharvest quality of the two cut rose cultivars, the flowers were treated either with $10\,\text{mL L}^{-1}$ NS, or a combination of NS and 0.1 mM AIB (NS+A), or a combination of NS+A and $1\,\mu\text{L L}^{-1}$ 1-MCP (NS+AM). Control (CON) flowers were placed in distilled water (DW). NS (Shanghai HuZheng Nano Tech Co., Ltd., Shanghai, China) and AIB (Sigma-Aldrich, Co., St. Louis, MO, USA) were diluted with DW to prepare the $10\,\text{mL L}^{-1}$ and 0.1 mM solutions. The generation of $1\,\mu\text{L L}^{-1}$ 1-MCP gas was conducted by breaking the sticks (FreshLong™, Ecoplants Co., Ltd., Yongin-si, Gyeonggi-do, Korea) four times and immediately placing in the treatment chambers at $25 \pm 1\,^\circ\text{C}$ under dark conditions. After 12 h of 1-MCP treatment, three cut rose flowers from 18 flowers in each treatment were sampled for gene expression analysis prior to ethylene exposure.

### 2.3. Ethylene Treatment

The ethylene treatment was conducted based on our previous study [1]. Briefly, all cut roses were sealed in the treatment chambers with $2\,\mu\text{L L}^{-1}$ ethylene at $25 \pm 1\,^\circ\text{C}$ under dark conditions. Two petri dishes containing 100 mL of 1000 mM NaOH were placed in the treatment chambers to neutralize carbon dioxide released by cut flower respiration during the ethylene treatment. After 24 h of ethylene exposure, six of the 15 flowers per treatment were placed in the controlled environment room at $25 \pm 1\,^\circ\text{C}$, RH of 50–55%, and with 12 h of light at a lux density of $30\,\mu\text{mol m}^{-2}\,\text{s}^{-1}$ to vase life assessment. The remaining nine cut roses from each treatment were used to conduct gene expression analysis after exposure to exogenous ethylene.

### 2.4. Measurements

#### 2.4.1. Anti-Bacterial Activity

To assess the antibacterial activity of the treatments, bacterial contamination was counted at the base of the flower stems after 4 days of ethylene exposure. Bacterial count methods were conducted based on the previous study [13]. The bacterial count was measured by counting the number of colonies formed on the plates after 2 days of incubation at $37 \pm 1\,^\circ\text{C}$.

#### 2.4.2. Flower Diameter (FD) and Water Relations

The effectiveness of the treatments on the two cut rose flowers was assessed by measuring changes in FD, fresh weight (FW), solution uptake (SU), and water balance (WB) at 9:30 a.m. daily. FD was measured by using a calliper (CD-20APX, Mitutoyo Corporation, Kanagawa, Kawasaki, Japan) FD was calculated as an average of the largest diameter and the diameter perpendicular to it. Separate FW of cut roses and SU were weighed during

vase periods, and SU and transpiration (TRANS) were calculated. TRANS was calculated using the formula $SU - FW_c - E$, where $FW_c$ is the change in FW of cut roses and E is the evaporation loss from a vase. The WB of cut roses was measured by subtracting TRANS from SU daily.

### 2.4.3. Maximal PSII Quantum Yield ($F_v/F_m$), Soluble Solids Content (SSC), and Chlorophyll Content

To evaluate the effectiveness of the treatments on the water stress status of the cut rose flowers, the chlorophyll fluorescence parameters, such as $F_0$ (minimal fluorescence), $F_m$ (maximal fluorescence), and $F_v$ (maximal variable fluorescence) in leaves were measured on day 4 in 'Matador' and day 9 in 'Dolcetto' by using the methods described in our previous study [13]. $F_0$ and $F_m$ was measured in full light-adapted conditions and in dark-adapted states using an Imaging Fluorometer (FluorCam 700MF, Photon Systems Instruments, Brno, Czech Republic). $F_v/F_m$, which reflects water stress status of cut flowers, was automatically calculated from $F_0$ and $F_m$.

The SSC (%) and chlorophyll content were also measured in the leaves of cut flowers on days 4 (in 'Matador') and 9 (in 'Dolcetto') of the vase periods, using the methods described in our previous study [22]. Briefly, the chlorophyll content of terminal leaflets was measured using a chlorophyll meter (SPAD-502Plus; Konica Minolta Sensing, Inc., Osaka, Japan) and SSC (%) of the uppermost leaves of the cut flowers was measured using a portable refractometer (PR-104, Atago, Tokyo, Japan).

### 2.4.4. Vase life (VL) and Senescence Symptoms

The VL of cut roses was determined as the time placing the cut flowers in the controlled environment room after ethylene exposure until a senescence symptom was reached. A VL assessment was conducted daily in accordance with the evaluation card for *Rosa* [23]. VL was considered to be terminated when the cut roses showed at least one of following senescence symptoms: wilting of petals (≥50% petal turgor loss), petal abscission (≥3 petal drop), bluing (≥50% blue petals), and leaf yellowing. No *Botrytis cinerea* infection was noted in this experiment.

### 2.5. RNA Isolation and Quantitative Real-Time PCR (qRT-PCR)

Total RNA was isolated from 200 mg of petals that were detached from cut roses on day 0 (before ethylene treatment) and on days 4, 6, and 10 after ethylene treatment using the GeneJET plant RNA purification Mini Kit (Thermo Fisher Scientific Baltics, Vilnius, Lithuania) with slight modifications of the manufacturer's protocol. cDNA was synthesized from 1 µg of the total RNA with an oligo $dT_{15}$ primer using a Power cDNA Synthesis Kit (INTRON Biotechnology, Inc., Seongnam, Gyeonggi-do, Korea). mRNA levels of ethylene biosynthesis (*RhACS2* and *RhACO1*) and receptor (*RhETR2*, *RhETR3*, and *RhETR5*) genes were determined using the StepOnePlus™ real-time PCR system (Applied Biosystems, Foster City, CA, USA). A fragment of *Rosa hybrida* actin1 (*RhACT1*) was used as the internal control to confirm the amount of the template RNA. qRT-PCR conditions for the detected genes have been described previously [9]. The primer sequences and accession number of the genes were listed in Table 1.

**Table 1.** Primer sequences used for qRT-PCR analyses in this study.

| Gene (Accession Number) | Forward Primer | Reverse Primer | Size |
|---|---|---|---|
| *RhACS2* (AY803737.1) | 5′-GCGAACAGGGGTACAACTTC-3′ | 5′-GGGTTTGAGGGGTTGGTAAT-3′ | 147 |
| *RhACO1* (AF441282.1) | 5′-CGTTCTACAACCCAGGCAAT-3′ | 5′-TTGAGGCCTGCATAGAGCTT-3′ | 130 |
| *RhETR2* (AF127220.1) | 5′-CTGCGTTAGAGCAGCAACTG-3′ | 5′-GGAATTCGGCGATATCTTCA-3′ | 131 |
| *RhETR3* (AY953392.1) | 5′-CCATGAGTTGAAAGGGAGGA-3′ | 5′-GGCTCACCAAAATCACCACT-3′ | 156 |
| *RhETR5* (AF441283.1) | 5′-TGTGTGGAGCGACACATCTT-3′ | 5′-TGAGGGCAGTAGCACATGAC-3′ | 120 |
| *RhACT1* (KC514918.1) | 5′-GTTCCCAGGAATCGCTGATA-3′ | 5′-ATCCTCCGATCCAAACACTG-3′ | 116 |

### 2.6. Experiment Design and Data Analysis

Vase life, MFD, SPAD, antibacterial activity, and water relations measurements were designed using six replicates for each treatment (one flower per replicate). qRT-PCR, $F_v/F_m$, and SSC analyses were conducted with three biological replicates. All data were shown as mean $\pm$ standard error (SE). A one-way analysis of variance (ANOVA) was conducted using SPSS 22.0 version (IBM, Somers, NY, USA). When significant effects were detected, a post-hoc pairwise comparison of group means was performed using Duncan's multiple range test (DMRT), with a significance level of $p < 0.05$.

## 3. Results

### 3.1. Senescence Symptoms and Longevity of Cut Roses

The senescence symptom that terminated VL was bluing (63%) followed by wilting (37%) in 'Matador' flowers, while wilting (60%), abscission (20%), and leaf yellowing (20%) were the main senescence symptoms in cut rose 'Dolcetto' (Figure 1A–D). All treatments significantly improved the visual appearance of the cut roses compared with that of the CON flowers, especially with respect to preventing petal wilting in 'Matador' and petal abscission and leaf yellowing in 'Dolcetto' (Figure 1A–D). On days 4 (in 'Matador') and 9 (in 'Dolcetto') of the vase life, most of the CON flowers showed senescence symptoms, whereas the cut flowers treated with NS, NS+A, and NS+AM retained good ornamental qualities (Figure 1A,B).

Compared with the CON flowers, all treatments significantly prolonged the postharvest lives of the cut roses of both cultivars (Figure 1E,F). However, cut roses treated with NS+AM exhibited the longest postharvest life (10.2 d in 'Matador' and 16.0 d in 'Dolcetto') (Figure 1E,F). These results showed that NS+AM was the most effective in delaying senescence and extending the postharvest life of cut roses in both cultivars after ethylene exposure.

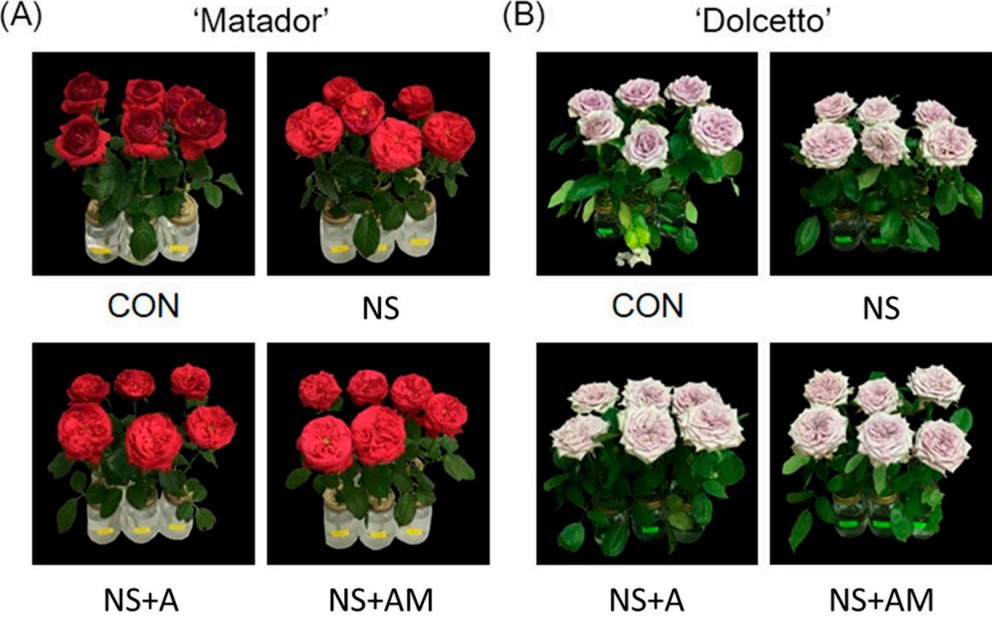

**Figure 1.** *Cont.*

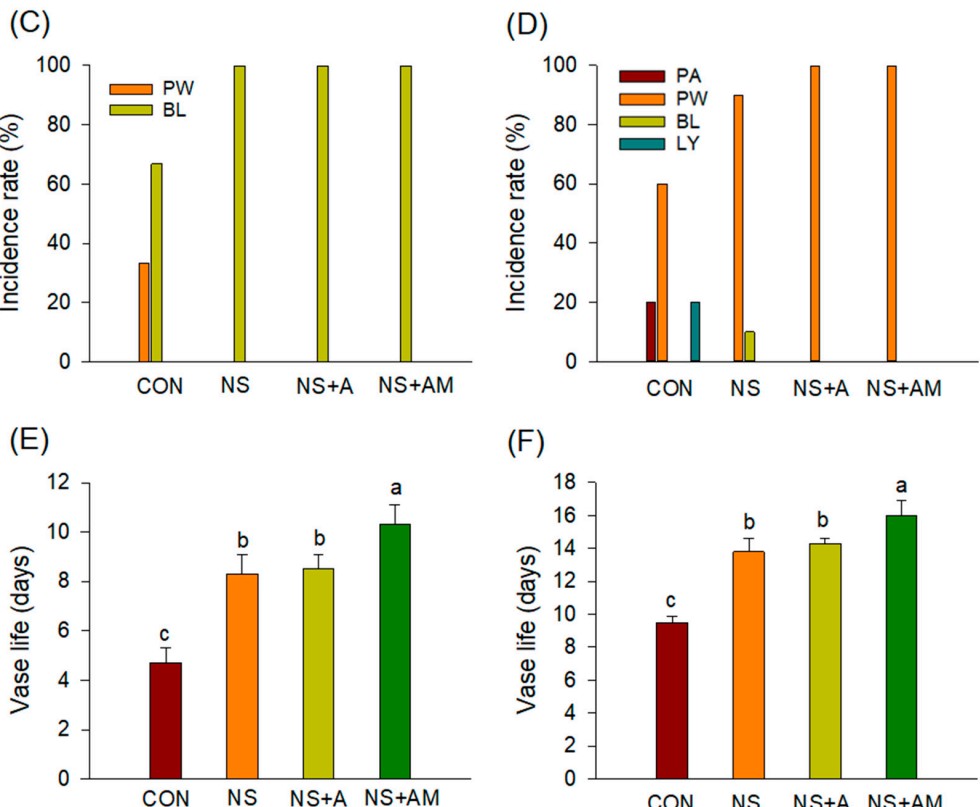

**Figure 1.** Ornamental quality performance, incidence rate of senescence symptoms, and vase life of 'Matador' (**A**,**C**,**E**) and 'Dolcetto' (**B**,**D**,**F**) cut roses. CON, control flowers were held in DW; NS, cut roses were treated with nano silver; NS+A, cut roses were treated with NS and AIB; NS+AM, cut roses were treated with NS, AIB, and 1-MCP. All cut roses were exposed to ethylene (2 μL L$^{-1}$) for 24 h at 25 ± 1 °C under dark conditions. PA, petal abscission; PW, petal wilting; BL, bluing; and LY, leaf yellowing. Different letters above bars show statistically significant differences among treatments at $p < 0.05$ based on DMRT. Vertical bars indicate SE of the means ($n = 6$).

### 3.2. Antibacterial Activity of the Treatments and Water Relations of Cut Flowers

The highest bacterial population densities were found on day 4 at the cut stem ends of both cultivars in the CON flowers (Figure 2). In contrast, only few bacterial colonies were developed on day 4 at the ends of the cut flower stems in the NS, NS+A, and NS+AM treatments (Figure 2).

The relative fresh weight (RFW) and SU rate are strongly related to the vase life of cut rose flowers. In the present study, changes in RFW and SU of both cultivars showed similar trends in all treatments. The RFW of CON flowers in 'Matador' and 'Dolcetto' declined on days 4 and day 5 of the vase periods, respectively (Figure 3A,B), the RFW of cut roses treated with NS, NS+A, and NS+AM was much higher than that of the CON flowers during the entire vase periods (Figure 3A,B). Among the treatments, the highest RFW of both cultivars was found in the NS+AM treatment (Figure 3A,B). A similar tendency was also observed in SU and WB among the treatments in both rose cultivars (Figure 3C–F).

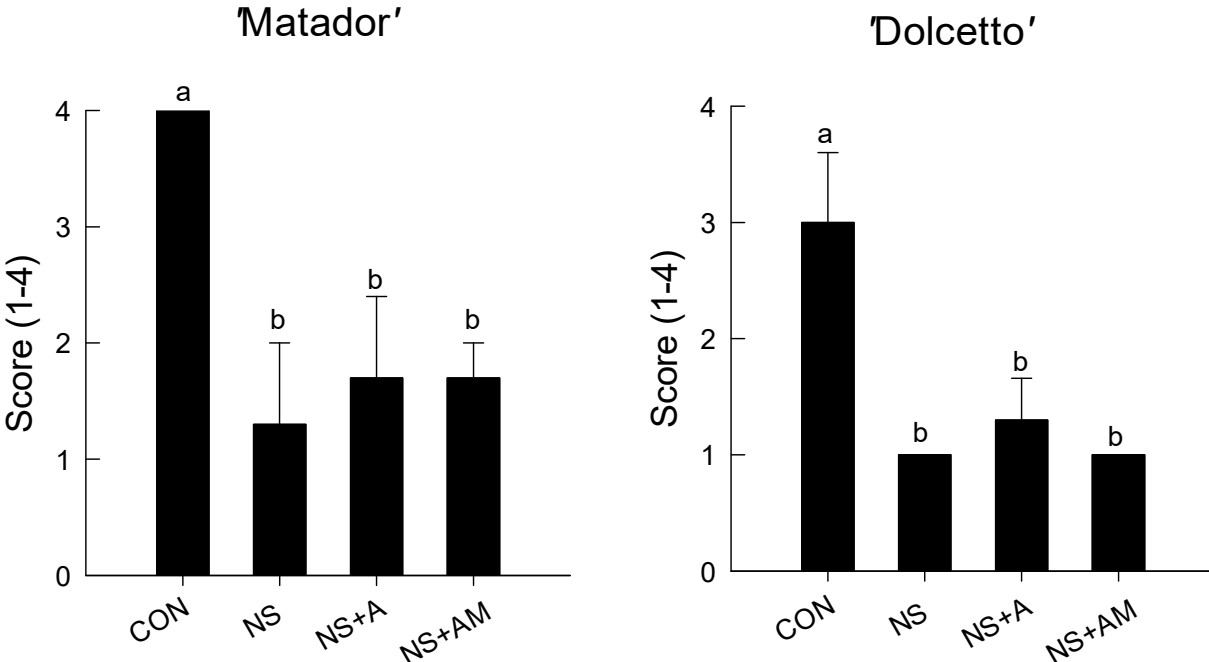

**Figure 2.** Bacterial population at the cut stem ends of 'Matador' and 'Dolcetto' rose cultivars after 4 days of the treatment. CON, control flowers were held in DW; NS, cut roses were treated with nano silver; NS+A, cut roses were treated with NS and AIB; NS+AM, cut roses were treated with NS, AIB, and 1-MCP. All cut roses were exposed to ethylene (2 μL L$^{-1}$) for 24 h at 25 $\pm$ 1 °C under dark conditions. Score: (1), 0 colony; (2), 1–10 colonies; (3), 10–50 colonies; and (4), >50 colonies. Different letters above bars show statistically significant differences among treatments at $p < 0.05$ based on DMRT. Vertical bars indicate SE of the means ($n = 6$).

### 3.3. FD and SSC of Cut Roses

The FD of cut flowers was measured daily to assess the effects of treatment solutions on flower opening. The largest maximum FD (MFD) was observed in NS+AM flowers of both rose cultivars, and NS and NS+A also significantly increased the MFD of the cut flowers, compared with that of CON flowers (Table 2). The SSC values were also measured in flower leaves to determine the relationship between flower opening and internal sucrose content of cut flowers, and the highest SSC of cut flowers in both cultivars was observed in the NS+AM treatment (Table 2).

**Table 2.** Effect of the treatments on flower opening, soluble solids content, $F_v/F_m$ ratios, and chlorophyll content of leaves of two rose cultivars.

| Cultivar | Treatment | MFD (% of Initial) | SSC (%) | $F_v/F_m$ | Chlorophyll Content (SPAD Value) |
|---|---|---|---|---|---|
| 'Matador' | CON | 114.8 $\pm$ 4.4 c $_X$ | 0.22 $\pm$ 0.00 c | 0.62 $\pm$ 0.03 b | 48.7 $\pm$ 0.2 b |
| | NS | 116.0 $\pm$ 3.3 b | 0.22 $\pm$ 0.00 c | 0.72 $\pm$ 0.02 ab | 49.9 $\pm$ 0.3 b |
| | NS+A | 118.6 $\pm$ 6.8 b | 0.30 $\pm$ 0.03 b | 0.74 $\pm$ 0.01 ab | 49.8 $\pm$ 0.4 b |
| | NS+AM | 129.7 $\pm$ 2.5 a | 0.33 $\pm$ 0.01 a | 0.75 $\pm$ 0.01 a | 51.5 $\pm$ 0.2 a |
| 'Dolcetto' | CON | 160.2 $\pm$ 4.6 c | 0.10 $\pm$ 0.00 c | 0.58 $\pm$ 0.06 c | 35.5 $\pm$ 5.8 c |
| | NS | 162.7 $\pm$ 6.7 c | 0.27 $\pm$ 0.06 b | 0.77 $\pm$ 0.06 ab | 48.4 $\pm$ 0.4 b |
| | NS+A | 169.9 $\pm$ 6.4 b | 0.27 $\pm$ 0.03 b | 0.76 $\pm$ 0.01 b | 48.3 $\pm$ 0.4 b |
| | NS+AM | 178.8 $\pm$ 5.2 a | 0.32 $\pm$ 0.06 a | 0.78 $\pm$ 0.01 a | 53.1 $\pm$ 4.7 a |

MFD (maximum flower diameter), SSC (soluble solids content), $F_v/F_m$ (the maximum quantum yield of dark-adapted PSII in leaves), and SPAD values were measured on day 4 in 'Matador' and on day 9 in 'Dolcetto'. CON, control flowers were held in DW; NS, cut roses were treated with nano silver; NS+A, cut roses were treated with NS and AIB; NS+AM, cut roses were treated with NS, AIB, and 1-MCP. All cut roses were exposed to ethylene (2 μL L$^{-1}$) for 24 h at 25 $\pm$ 1 °C under dark conditions. $_X$ Different letters (a–c) among treatments indicate statistically significant differences at $p < 0.05$ based on DMRT ($n = 6$ for MFD and SPAD and 3 for SSC and $F_v/F_m$).

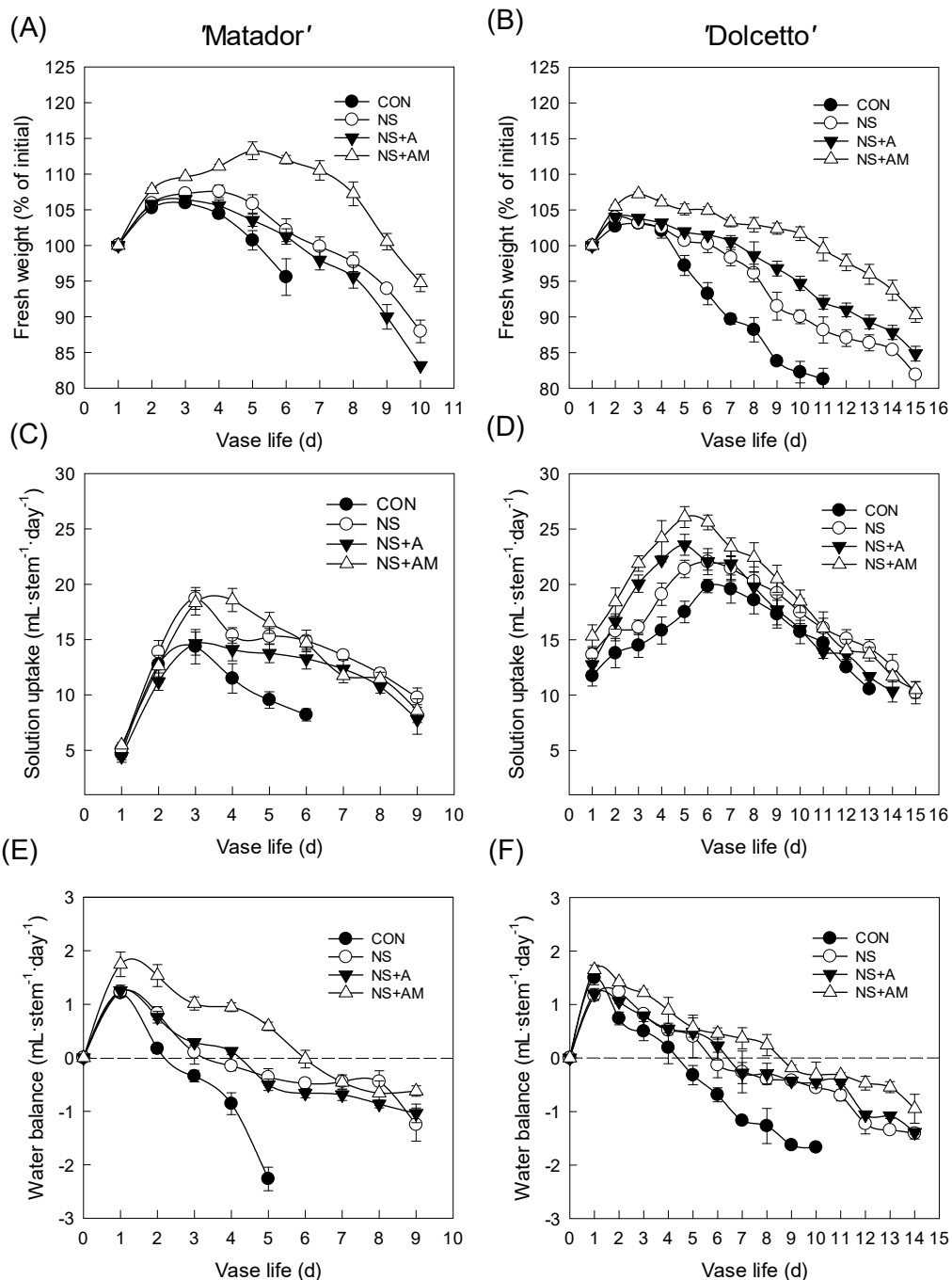

**Figure 3.** Changes in relative fresh weight, solution absorption, and water balance in 'Matador' (**A**,**C**,**E**) and 'Dolcetto' (**B**,**D**,**F**) cut roses. CON, control flowers were held in DW; NS, cut roses were treated with nano silver; NS+A, cut roses were treated with NS and AIB; NS+AM, cut roses were treated with NS, AIB, and 1-MCP. All cut roses were exposed to ethylene (2 µL L$^{-1}$) for 24 h at $25 \pm 1$ °C under dark conditions. Vertical bars show SE of the mean (*n* = 6).

### 3.4. $F_v/F_m$ and Chlorophyll Content (SPAD)

The water stress status of cut flowers is strongly associated with the $F_v/F_m$ ratios in leaves. Plants often exhibit decreased $F_v/F_m$ values in response to the water stress conditions [24]. In this study, the greatest reduction in $F_v/F_m$ values occurred in the leaves of CON flowers after ethylene treatment, whereas $F_v/F_m$ values of both cultivars were effectively maintained during vase life by NS, NS+A, and NS+AM treatments (Table 2). As expected, the $F_v/F_m$ values were highest in flowers treated with NS+AM (Table 2),

whereas the values decreased in CON flowers in accordance with the flower wilting and leaf yellowing after 4 days (in 'Matador') and 9 days (in 'Dolcetto') of ethylene exposure (Table 2 and Figure 1A,B).

Leaf chlorophyll measured by SPAD was the lowest in CON flowers, while degradation in leaf chlorophyll of both cultivars was effectively delayed in all treatments (Table 2). The highest SPAD value was observed for NS+AM flowers (Table 2). These results indicate that leaf SPAD and $F_v/F_m$ values are strongly associated with flower wilting and leaf yellowing in cut roses.

### 3.5. Changes in mRNA Levels of Ethylene Biosynthesis and Receptor Genes

Based on our previous findings [1,9], the mRNA levels of two ethylene biosynthesis genes (*RhACS2* and *RhACO1*) and three receptor isoforms (*RhETR2*, *RhETR3*, and *RhETR5*) were detected in rose petals during vase periods to examine the effects of treatments on ethylene response of the cut flowers. The mRNA levels of *RhACS2* and *RhACO1* in the CON flowers of both cultivars were significantly increased after ethylene exposure (Figure 4A–D). The NS, NS+A, and NS+AM treatments effectively suppressed the ethylene-inducible increase in the mRNA levels of *RhACS2* in both cultivars (Figure 4A,B). Treatment with NS and NS+A inhibited *RhACO1* transcript levels in rose petals on day 4 in 'Matador' and days 4 and 6 in 'Dolcetto', whereas NS+AM effectively suppressed the increase in *RhACO1* expression in 'Matador' during vase periods (Figure 4C,D). In contrast, in 'Dolcetto', the transcript levels of *RhACO1* in petals of NS+AM flowers were increased on day 10 of the vase life (Figure 4D).

Unlike the ethylene biosynthesis genes, the mRNA levels of three ethylene receptor genes were the lowest in petals of CON flowers after the ethylene treatment (Figure 5). However, the ethylene-inducible decrease in the receptor genes in rose petals was effectively repressed in the NS, NS+A, and NS+AM treatments on days 4 and 6 in 'Matador' (Figure 5A,C,E). Interestingly, the degradation of three ethylene receptor genes in the petals of cut roses was most strongly suppressed in the NS+AM treatment (Figure 5A,C,E). In 'Dolcetto', only NS+AM treatment reduced the degradation of the three receptor isoforms in rose petals at early stage (days 4 and 6) of the vase life (Figure 5B,D,F). It is likely that the increase in *RhACO1* mRNA accumulation in the later stage of NS+AM flowers (day 10) induced the degradation of ethylene receptor genes (Figure 5B,D,F). These findings indicate that the simultaneous inhibition of ethylene biosynthesis and action is much more effective in preventing an ethylene response in ethylene-sensitive cultivars compared to ethylene-insensitive cultivars.

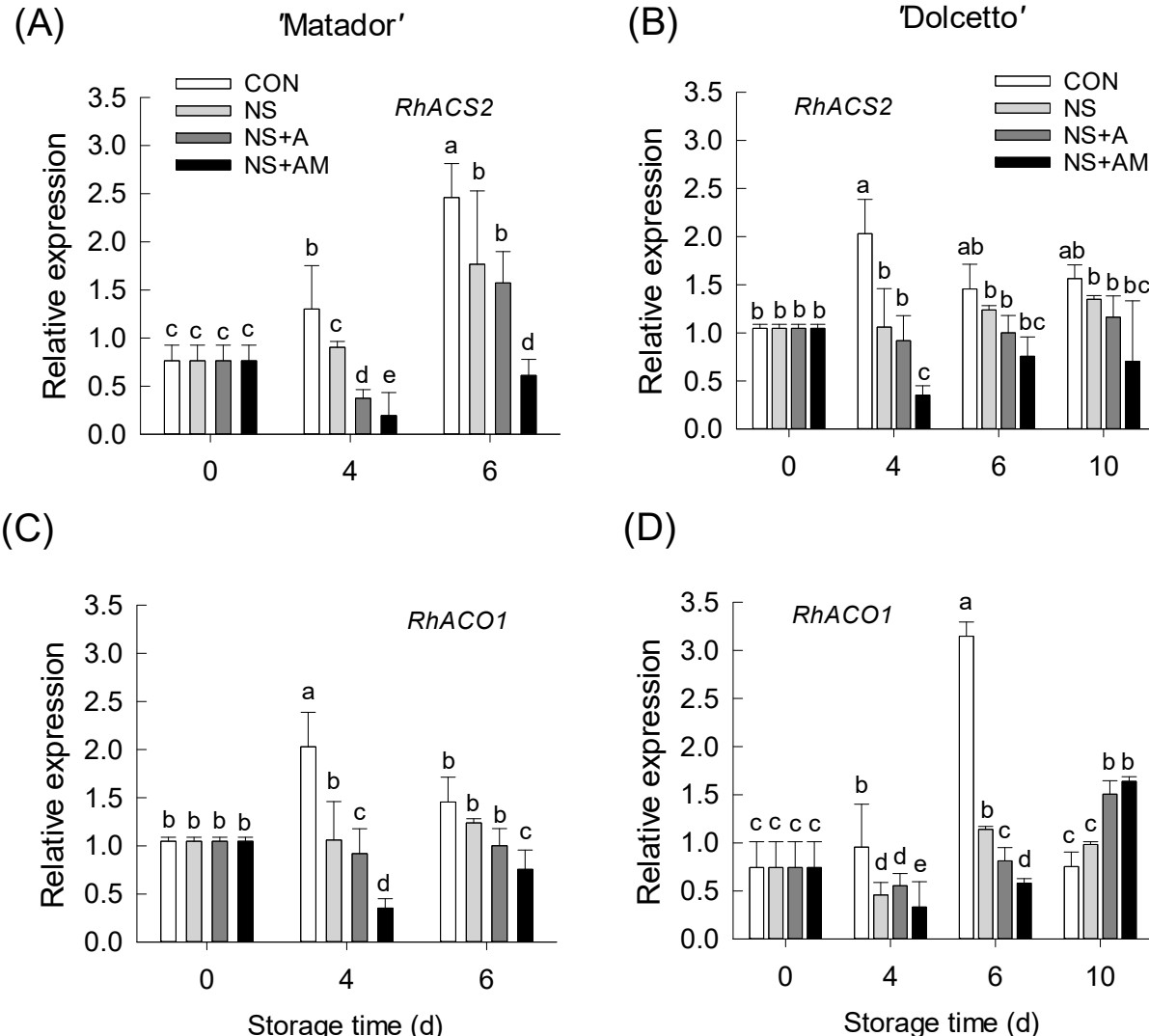

**Figure 4.** Expression levels of ethylene biosynthesis genes (*RhACS2* and *RhACO1*) in 'Matador' (**A**,**C**) and 'Dolcetto' (**B**,**D**) cut roses. mRNA levels of *RhACS2* and *RhACO1* were detected in petals on day 0 (before ethylene treatment) and on days 4, 6, and 10 after ethylene treatment. CON, control flowers were held in DW; NS, cut roses were treated with nano silver; NS+A, cut roses were treated with NS and AIB; NS+AM, cut roses were treated with NS, AIB, and 1-MCP. All cut roses were exposed to ethylene (2 μL L$^{-1}$) for 24 h at 25 ± 1 °C under dark conditions. Different letters above bars show statistically significant differences among treatments at $p < 0.05$ based on DMRT. Vertical bars indicate SE of the means ($n = 3$).

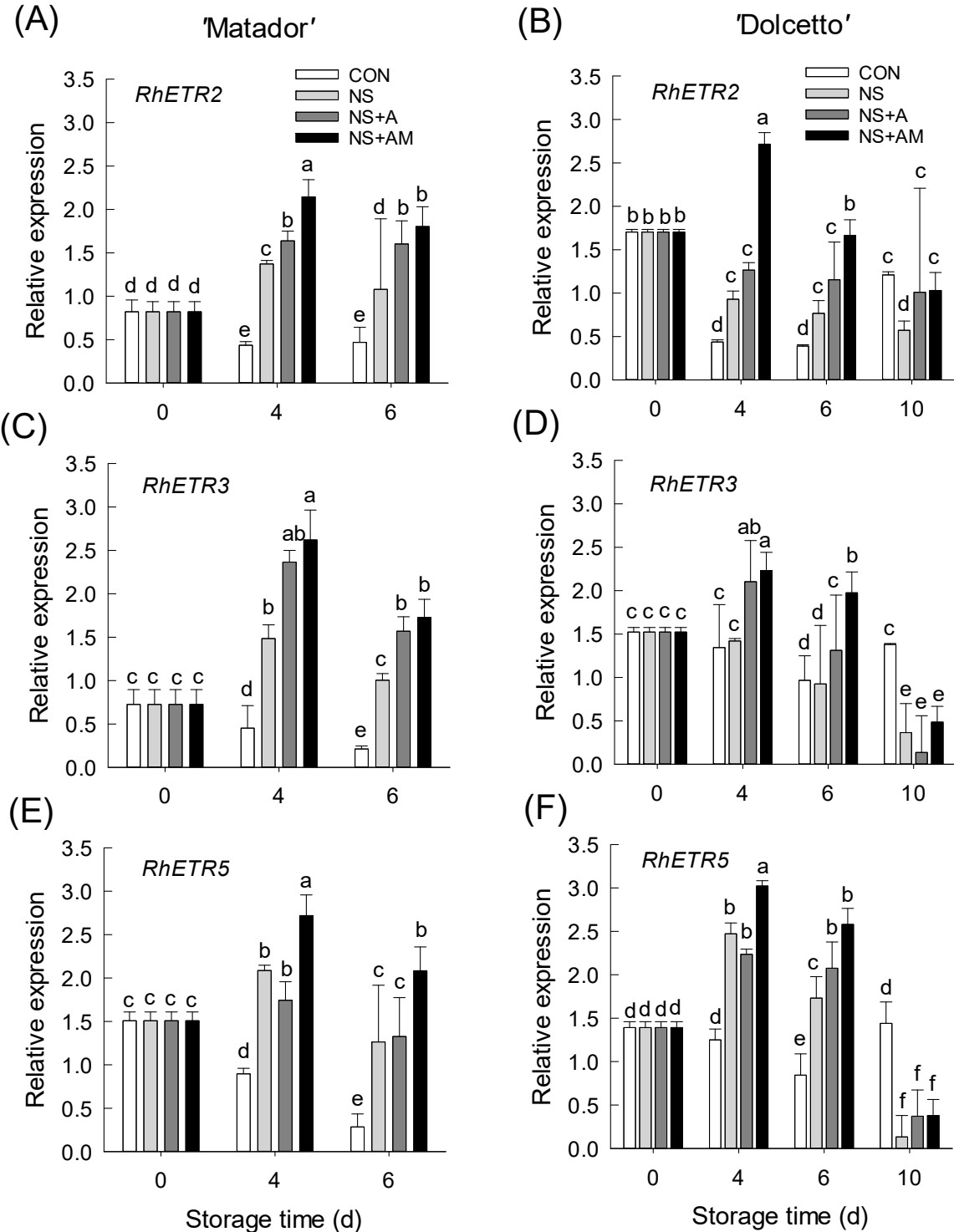

**Figure 5.** Expression levels of *RhETR2*, *RhETR3*, and *RhETR5* in 'Matador' (**A,C,E**) and 'Dolcetto' (**B,D,F**) cut roses. mRNA levels of three receptor genes were detected in petals on day 0 (before ethylene treatment) and on days 4, 6, and 10 after ethylene treatment. CON, control flowers were held in DW; NS, cut roses were treated with nano silver; NS+A, cut roses were treated with NS and AIB; NS+AM, cut roses were treated with NS, AIB, and 1-MCP. All cut roses were exposed to ethylene (2 μL L$^{-1}$) for 24 h at 25 ± 1 °C under dark conditions. Different letters above bars show statistically significant differences among treatments at *p* < 0.05 based on DMRT. Vertical bars indicate SE of the means (*n* = 3).

## 4. Discussion

Cut roses lose their ornamental value due to flower senescence. Ethylene and water stress relating to the growth of bacteria in vase solution can cause early flower senescence symptoms such as flower wilting, abscission, or stem bending in many cultivars of cut rose flowers [1,13,18]. Bacterial growth in the vase solutions causes the blocking of vessels, which results in a postharvest water deficit and a lack of nutrition support in the flower stems [3,4]. Antimicrobial agents enhance water absorption by suppressing bacterial growth in vase solutions and within the stem vessels, thereby contributing to high petal turgor [25]. However, in ethylene-sensitive flowers, treatment with the biocides is less effective in prolonging vase life because the flowers can be damaged by ethylene in the early stages of vase life [1]. Thus, the most effective method to prolong the postharvest life of cut roses is through reduction in water stress and ethylene damages using preservative solutions.

In the current study, all treatments had positive effects on the postharvest quality and vase life of cut rose flowers. The use of NS+AM significantly extended the postharvest life of cut roses by 217% (5.5 d) in the ethylene-sensitive cultivar ('Matador') and by 168.4% (6.5 d) in the ethylene-insensitive cultivar ('Dolcetto') compared with CON flowers. The considerable extension in the longevity of cut flowers treated with NS+AM probably resulted from the combined effective of suppressing both water stress and ethylene damages.

Bacterial growth in the vase solutions has been shown to lead to xylem plugging, which consequently restricts the water flux in the stems of cut roses and results in a short lifespan [3]. In this study, all the treatments effectively inhibited bacterial proliferation at the cut stem ends of roses during vase periods. This result can be explained by the function of NS as a strong antibacterial agent [17]. NS particles have recently emerged as safe antimicrobial agents, and they have a stronger microorganism inhibiting effect than silver ions when used in other oxidation compounds, such as STS or $AgNO_3$, because of their high surface area to volume ratio [16,17]. Compared with the other treatments, NS+AM treatment enhanced SU, which resulted in a relatively longer retention time of positive WB and initial FW with both rose cultivars. This result is consistent with that of a previous study in which treatment with NS increased water absorption in the cut flower tissues of gerberas, resulting in decreased stem bending and an extension of their lifespan [18]. In addition, when water stress was completely suppressed by the NS+AM treatment, flower wilting of the cut rose flowers was inhibited, which resulted in relatively higher $F_v/F_m$ in both rose cultivars than in the other treatments. Furthermore, the high leaf chlorophyll content in NS+AM flowers contributed to the increased $F_v/F_m$ values in both rose cultivars.

A previous study revealed that the SSC is often depleted during senescence in both intact and cut flowers [25]. It is interesting to note that the NS+AM treatment increased the SSC in the cut flowers of both cultivars. The SSC levels of NS+AM flowers were closely related to the increased flower diameters by day 4 in 'Matador' and by day 9 in 'Dolcetto'. These results are consistent with previous observations that indicated an internal sucrose content provides the energy required for flower opening during vase periods [22,26].

In the ethylene-sensitive cultivar ('Matador'), flower senescence in the CON group was strongly correlated with the increased mRNA levels of ethylene biosynthesis genes and the decreased expression of three receptor isoforms after exposure to ethylene. Compared with the other treatments, NS+AM effectively prevented early flower senescence during the postharvest life of cut roses. Previous observations have shown that the close relationship between ACS and ethylene sensitivity levels and that the distinct ethylene sensitivity of cut flowers may be related to differences in receptor levels [27–30]. In this study, it is likely that the simultaneous treatment of AIB and 1-MCP with NS directly suppressed cut rose responsiveness to exogenous ethylene during vase life by inhibiting ACS activity and preventing ethylene binding to the receptors, thereby delaying the senescence of cut flowers. Previous studies have also showed that $Ag^{+1}$ ions released from NS effectively inhibit ethylene perception by replacing the cofactor $Cu^{+2}$ for the ethylene receptors, and they could interfere with ethylene biosynthesis by reducing the expression of *ACS* and

*ACO* genes [31,32]. NS has been found to effectively decrease the *mRNA* levels of ethylene biosynthesis genes and ethylene responses in ethylene-sensitive carnations, roses, and *Arabidopsis* seedlings [19,21,32], and NS has been found to significantly prevented the degradation of ethylene receptors and the downstream protein kinase, which results in the inhibition of the ethylene responses in rose petals [19]. Our results imply that NS contributes to the suppression of ethylene responses in the petals of cut roses.

In contrast, the flower senescence in ethylene-insensitive cultivars less affected by exogenous ethylene, and the relationship between ethylene and petal senescence in the ethylene-insensitive flowers also is unclear [1,33]. In ethylene-insensitive cultivar ('Dolcetto'), NS+AM significantly inhibited the increased expression of *RhACO1* and the reduction in three ethylene receptor genes only at the early stage (days 4 and 6) of the vase periods. This result indicates that ethylene response in rose petals was not completely prevented by the combined addition of AIB and 1-MCP in ethylene-insensitive cultivars during the entire vase period. Therefore, flower senescence in 'Dolcetto' may be more closely related to water stress than ethylene damage. These results are also consistent with those of our previous study, which found that the vase life of ethylene-insensitive flowers was strongly associated with water relation failure [1]. The results of the current study suggest that the use of NS+AM could play an important role in improving the water relations of the cut rose 'Dolcetto' rather than suppressing its ethylene response, which would extend its vase life. Physiological processes of reduced solution uptake and wilting of rose petals are likely related to the expression of genes related to water transporter in cut roses, such as aquaporin (*RhPIPs* or *RhTIPs*), carbohydrate metabolism, and ion transporters [34,35]. Thus, NS+AM may reduce water stress and early wilting of petals in cut 'Dolcetto' roses through regulating the expression of water transporter genes. However, further molecular analyses are necessary to elucidate the role of NS+AM in water transport regulation in cut rose of ethylene insensitive cultivars.

Overall, our results show that the combination of ethylene biosynthesis and binding inhibitors with antibacterial agent decreases ethylene damage and water stress of both ethylene-sensitive and ethylene-insensitive rose cultivars, which resulted in longer vase life of both. The results of this study will facilitate the development of new preservatives that could prolong the vase life of cut flowers from a wide range of flower species.

**Author Contributions:** S.T.T.H. performed the experiment, data collection, and analysis, and wrote the draft of the manuscript. B.-C.I. designed the study, contributed to the data analysis, wrote the manuscript, and supervised the project. All authors have read and agreed to the published version of the manuscript.

**Funding:** This study was funded by a research grant (PJ0150142022) from the Rural Development Administration of Korea (RDA) and the Basic Research Program (NRF-2021R1I1A3A04037108) through the National Research Foundation of Korea (NRF) funded by the Ministry of Education, Science and Technology.

**Institutional Review Board Statement:** Not applicable.

**Informed Consent Statement:** Not applicable.

**Data Availability Statement:** All data generated and analyzed during this study are included in this published article.

**Acknowledgments:** We thank the members of the Postharvest Physiology Lab for their assistance during the measurements.

**Conflicts of Interest:** The authors have no conflicts of interest to declare.

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
