# Peer review of "Combined Nano Silver, α-Aminoisobutyric Acid, and 1-Methylcyclopropene Treatment Delays the Senescence of Cut Roses with Different Ethylene Sensitivities"

_horticulturae, doi:10.3390/horticulturae8060482_

Round 1

Reviewer 1 Report

This research described three treatments to effectively extend vase life of two rose cultivars (ethylene-sensitive and ethylene-insensitive). NS, AIB and 1-MCP were all reported to retard petal senescence and extend vase life in many flowers. Multiple experiments were conducted to describe the physiological phenotypes and gene expressions to explore how NS, NS+A and NS+AM can retard petal senescence and extend vase life. It is indicated that NS+AM most effectively suppressed the ethylene response and water stress of both cultivars, and NS+AM can supposed be a new preservative. However, there are still many questions:

(1) Figure 1C, the senescence symptom that terminated vase life in CON Matador is Bluing and petal wilting, while in NS, NS+A and NS+AM Matador is only Bluing, so I think there should be some else parameters about the bluing rate to describe why NS, NS+A and NS+AM can extend the vase life.

(2) Dolcetto is ethylene insensitive, and petal wilting is the most factor to terminate vase life, the gene expression related to water transporter should be detected to better interpret why NS, NS+A and NS+AM can extend the vase life.

(3) Figure 4A, CON Matador is totally senescence at day 6 after ethylene treatment, ACS2 is still highly expressed, this result should be double checked.

(4) In material and method 4.2, only 18 flowers were analyzed for gene expression and physiological phenotypes, it is not enough to explore effective preservative. If there is multiple biological repetition or batch repeat, it should be described in detail.

(5) Do the authors try any other ratio and concentration of NS, AIB and 1-MCP, there might be more effective preservative.

Reviewer 2 Report

MS titled “Combined Nano Silver, AIB, and 1-MCP Treatment Delays the Senescence of Cut Roses with Different Ethylene Sensitivities”. The work is interesting and contains some novel information. In my opinion, it needs some changes before reconsideration for possible publication. It is suggested to write full terms in the title and acronyms should be avoided. The abstract is well written and easy to understand but the concluding sentences should be further clarified. Keywords should be different from the title. The results are easy to understand and nicely written but I strongly suggest to significantly improve the discussion part of the manuscript. The conclusion should also be improved by omitting the repeated results in it. Please supplement figure legends as “storage time (d)” not time (d). Please improve the table 1 title caption.

Reviewer 3 Report

This is a well written manuscript that falls into the scope of the Journal. I a have only minor recommendations amended on the attached pdf. 

Round 2

Reviewer 1 Report

The authors have addressed all my concerns.